# PeerJ

# Automatic large-scale classification of bird sounds is strongly improved by unsupervised feature learning

Dan Stowell and Mark D. Plumbley

Centre for Digital Music, Queen Mary University of London, UK

## ABSTRACT

Automatic species classification of birds from their sound is a computational tool of increasing importance in ecology, conservation monitoring and vocal communication studies. To make classification useful in practice, it is crucial to improve its accuracy while ensuring that it can run at big data scales. Many approaches use acoustic measures based on spectrogram-type data, such as the Mel-frequency cepstral coefficient (MFCC) features which represent a manually-designed summary of spectral information. However, recent work in machine learning has demonstrated that features learnt automatically from data can often outperform manually-designed feature transforms. Feature learning can be performed at large scale and "unsupervised", meaning it requires no manual data labelling, yet it can improve performance on "supervised" tasks such as classification. In this work we introduce a technique for feature learning from large volumes of bird sound recordings, inspired by techniques that have proven useful in other domains. We experimentally compare twelve different feature representations derived from the Mel spectrum (of which six use this technique), using four large and diverse databases of bird vocalisations, classified using a random forest classifier. We demonstrate that in our classification tasks, MFCCs can often lead to worse performance than the raw Mel spectral data from which they are derived. Conversely, we demonstrate that unsupervised feature learning provides a substantial boost over MFCCs and Mel spectra without adding computational complexity after the model has been trained. The boost is particularly notable for single-label classification tasks at large scale. The spectro-temporal activations learned through our procedure resemble spectro-temporal receptive fields calculated from avian primary auditory forebrain. However, for one of our datasets, which contains substantial audio data but few annotations, increased performance is not discernible. We study the interaction between dataset characteristics and choice of feature representation through further empirical analysis.

Corresponding author
Dan Stowell,
dan.stowell@qmul.ac.uk

## INTRODUCTION

Automatic species classification of birds from their sounds has many potential applications in conservation, ecology and archival (*Laiolo, 2010*; *Digby et al., 2013*; *Ranft, 2004*). However, to be useful it must work with high accuracy across large numbers of possible

species, on noisy outdoor recordings and at big data scales. The ability to scale to big data is crucial: remote monitoring stations can generate huge volumes of audio recordings (*Aide et al., 2013*), and audio archives contain large volumes of audio, much of it without detailed labelling. For example the British Library Sound Archive holds over 100,000 recordings of bird sound in digital format, from various sources (*Ranft, 2004*). Big data scales also imply that methods must work without manual intervention, in particular without manual segmentation of recordings into song syllables, or into vocal/silent sections. The lack of segmentation is a pertinent issue for both remote monitoring and archive collections, since many species of bird may be audible for only a minority of the recorded time, and therefore much of the audio will contain irrelevant information.

The task of classifying bird sounds by species has been studied by various authors, at least as far back as *McIlraith & Card (1997)*. (See *Stowell & Plumbley (2010)* for a survey.) Many of the early studies used small datasets, often noise-free and/or manually-segmented and with a small number of species, so their practical applicability for ecological applications is unclear. More recent studies have fewer such limitations, and introduce useful methods customised to the task (*Lakshminarayanan, Raich & Fern, 2009*; *Damoulas et al., 2010*; *Briggs et al., 2012*). However, there remain questions of scalability, due to the computational intensity of algorithms or to procedures such as all-pairs comparisons which cannot be applied to arbitrarily large datasets without modification (*Damoulas et al., 2010*).

In addition to noise-robustness and scalability issues, one further issue is the number of species considered by a classifier: certain classification systems may be developed to distinguish among ten or twenty species, but in many parts of the world there are hundreds of species that might be heard (*Ballmer et al., 2013*). Further, typical recordings in the wild contain sounds from more than one bird, and so it is advantageous to consider the task as a *multi-label* task, in which the classifier must return not one label but a set of labels representing all species that are present (*Briggs et al., 2012*).

One recent research project (named "SABIOD") has provided a valuable stimulus to the research community by conducting classification challenges evaluated on large datasets of bird sounds collected in the wild, and with large numbers of species to recognise (*Glotin et al., 2013*; *Fodor, 2013*; *Goëau et al., 2014*). The research reported in this paper benefits from the datasets made available through that project, as well as other datasets, to evaluate bird sound classification suitable for large-scale practical deployments.

Some previous work has compared the performance of different classification algorithms for the task (*Acevedo et al., 2009*; *Briggs, Raich & Fern, 2009*). In the present work, we instead use a standard but powerful classification algorithm, and focus on the choice of audio features used as input data. We introduce the concept of *feature learning* which has been applied in other machine learning domains, and show that in most cases it can lead the classifier to strongly outperform those using common MFCC and Mel spectrum features. We also evaluate the role of other aspects such as noise reduction in the feature preprocessing; however, the strongest effect of the parameters we study comes from replacing MFCCs with learned features.

In the following, we use four large and diverse birdsong datasets with varying characteristics to evaluate classifier performance. Overall, feature learning enables a classifier to perform very strongly on large datasets with large numbers of species, and achieves this boost with very little computational cost after the training step. Three of the four datasets demonstrate clearly the boost attained through feature learning, attaining very strong performance in both single-label and multi-label classification tasks. One dataset, consisting of long dawn-chorus recordings with a substantial amount of audio but few annotations, does not yield a significant benefit from the improved feature representation. We explore the reasons for this in follow-up experiments in which the training data is augmented or substituted with other data. Before describing our experiment, however, we discuss the use of spectral features and feature learning for audio classification.

## Spectral features and feature learning

Raw audio data is not generally suitable input to a classification algorithm: even if the audio inputs were constrained to a fixed duration (so that the data dimensionality was constant), the dimensionality of an audio signal (considered as a vector) would be extremely large, and would not represent sound in such a way that perceptually similar sounds would generally be near neighbours in the vector space. Hence audio data is usually converted to a spectrogram-like representation before processing, i.e., the magnitudes of short-time Fourier transformed (STFT) frames of audio, often around 10 ms duration per frame. (Alternatives to STFT which have been considered for bird sound classification include linear prediction (*Fox, 2008*), wavelets (*Selin, Turunen & Tanttu, 2007*) and chirplets (*Stowell & Plumbley, in press*)). An STFT spectrum indicates the energy present across a linear range of frequencies. This linear range might not reflect the perceptual range of a listener, and/or the range of frequencies at which the signal carries information content, so it is common to transform the frequency axis to a more perceptual scale, such as the Mel scale originally intended to represent the approximately logarithmic frequency sensitivity of human hearing. This also reduces the dimensionality of the spectrum, but even the Mel spectrum has traditionally been considered rather high-dimensional for automatic analysis. A convention, originating from speech processing, is to transform the Mel spectrum using a cepstral analysis and then to keep the lower coefficients (e.g., the first 13) which typically contain most of the energy (*Davis & Mermelstein, 1980*). These coefficients, the Mel frequency cepstral coefficients (MFCCs), became widespread in applications of machine learning to audio, including bird vocalisations (*Stowell & Plumbley, 2010*).

MFCCs have some advantages, including that the feature values are approximately decorrelated from each other, and they give a substantially dimension-reduced summary of spectral data. Dimension reduction is advantageous for manual inspection of data, and also for use in systems that cannot cope with high-dimensional data. However, as we will see, modern classification algorithms can cope very well with high-dimensional data, and dimension reduction always reduces the amount of information that can be made available to later processing, risking discarding information that a classifier could have

used. Further, there is little reason to suspect that MFCCs should capture information optimal for bird species identification: they were designed to represent human speech, yet humans and birds differ in their use of the spectrum both perceptually and for production. MFCCs aside, one could use raw (Mel-)spectra as input to a classifier, or one could design a new transformation of the spectral data that would tailor the representation to the subject matter. Rather than designing a new representation manually, we consider automatic feature learning.

The topic of feature learning (or representation learning, dictionary learning) has been considered from many perspectives within the realm of statistical signal processing (*Bengio, Courville & Vincent, 2013*; *Jafari & Plumbley, 2011*; *Coates & Ng, 2012*; *Dieleman & Schrauwen, 2013*) . The general aim is for an algorithm to learn some transformation that, when applied to data, improves performance on tasks such as sparse coding, signal compression or classification. This procedure may be performed in a "supervised" manner, meaning it is supplied with data as well as some side information about the downstream task (e.g., class labels), or "unsupervised", operating on a dataset but with no information about the downstream task. A simple example that can be considered to be unsupervised feature learning is principal components analysis (PCA): applied to a dataset, PCA chooses a linear projection which ensures that the dimensions of the transformed data are decorrelated (*Bengio, Courville & Vincent, 2013*). It therefore creates a new feature set, without reference to any particular downstream use of the features, simply operating on the basis of qualities inherent in the data.

Recent work in machine learning has shown that unsupervised feature learning can lead to representations that perform very strongly in classification tasks, despite their ignorance of training data labels that may be available (*Coates & Ng, 2012*; *Bengio, Courville & Vincent, 2013*). This rather surprising outcome suggests that feature learning methods emphasise patterns in the data that turn out to have semantic relevance, patterns that are not already made explicit in the basic feature processing such as STFT. A second surprising aspect is that such representations often perform the opposite of feature reduction, increasing the dimensionality of the problem without adding any new information: a deterministic transformation from one feature space to a higher-dimensional feature space cannot, in an information-theoretic sense, add any information that is not present in the original space. However, such a transformation can help to reveal the manifold structure that may be present in the data (*Olshausen & Field, 2004*) . Neural networks, both in machine implemetations and in animals, perform such a dimension expansion in cases where one layer of neurons is connected as input to a larger layer of neurons (*Olshausen & Field, 2004*).

In our study, however, we will not use a feature learning procedure intended to parallel a biological process. Instead, we use *spherical k-means*, a simple and highly scalable modification of the classic k-means algorithm (*Coates & Ng, 2012*; *Dieleman & Schrauwen, 2013*). We perform a further adaptation of the algorithm to ensure that it can run in streaming fashion across large audio datasets, to be described in 'Materials and Methods'.

Birdsong often contains rapid temporal modulations, and this information should be useful for identifying species-specific characteristics (*Stowell & Plumbley, in press*). From this perspective, a useful aspect of feature learning is that it can be applied not only to single spectral frames, but to short sequences (or "patches") of a few frames. The representation can then reflect not only characteristics of instantaneous frequency patterns in the input data, but characteristics of frequencies and their short-term modulations, such as chirps sweeping upwards or downwards. This bears some analogy with the "delta-MFCC" features sometimes used by taking the first difference in the time series of MFCCs, but is more flexible since it can represent amplitude modulations, frequency modulations, and correlated modulations of both sorts (cf. *Stowell & Plumbley, in press*). In our study we tested variants of feature learning with different temporal structures: either considering one frame at a time (which does not capture modulation), multiple frames at a time, or a variant with two layers of feature learning, which captures modulation across two timescales.

## MATERIALS AND METHODS

Our primary experiment evaluated automatic species classification separately across four different datasets of bird sound. For each dataset we trained and tested a random forest classifier (*Breiman, 2001*), while systematically varying the following configuration parameters to determine their effect on performance:

- Choice of features (MFCCs, Mel spectra, or learned features) and their summarisation over time (mean and standard deviation, maximum, or modulation coefficients);
- Whether or not to apply noise reduction to audio spectra as a pre-processing step;
- Decision windowing: whether to treat the full-length audio as a single unit for training/testing purposes, or whether to divide it into shorter-duration windows (1, 5 or 60 s);
- How to produce an overall decision when using decision windowing (via the mean or the maximum of the probabilities);
- Classifier configuration: the same random forest classifier tested in single-label, multilabel or binary-relevance setting.

We will say more about the configuration parameters below. Each of the above choices was tested in all combinations (a "grid search" over possible configurations) for each of our datasets separately, thus providing a rigorous search over a vast number of classifier settings, in up to 384 individual crossvalidated classification tests for each dataset.

In follow-up experiments we explored some further issues and their effect on species recognition:

- We separated out two aspects of our different feature sets—their dimensionality and their intrinsic character—by projecting the feature data to the same fixed dimensionality, and then re-testing with these;
- We tested the effect of data expansion, by training on the union of two datasets;

**Table 1** Summary of bird sound datasets used.

| Dataset | Location | Items | Total duration | Mean duration | Classes | Labelling |
|---------|----------|-------|----------------|---------------|---------|-----------|
| *nips4b* | France | 687 | 0.8 h (125k frames) | 4 s | 87 | Multilabel |
| *xccoverbl* | UK/Europe | 264 | 4.9 h (763k frames) | 67 s | 88 | Single-label |
| *bldawn* | UK | 60 | 7.8 h (1.2M frames) | 468 s | 77 | Multilabel |
| *lifeclef2014* | Brazil | 9688 | 77.8 h (12M frames) | 29 s | 501 | Single-label |

- We tested the effect of cross-condition training, by training on one dataset and testing with a different dataset.

## Datasets

We gathered four datasets, each representing a large amount of audio data and a large number of species to classify (Table 1). Two of the datasets (*nips4b* and *lifeclef2014*) consist of the publicly-released training data from bird classification challenges organised by the SABIOD project (*Glotin et al., 2013*; *Goëau et al., 2014*). The *nips4b* dataset is multilabel (median 1 species per recording, range 0–6); the *lifeclef2014* dataset is single-label but much larger. (Some of the data in *lifeclef2014* includes annotations of "background species" which could be used alongside the primary annotation to construct a multilabel task; we did not do this.) Note that we only use the publicly-released training data from those challenges, and not any private test data, and so our evaluation will be similar in nature to their final results but not precisely comparable. For evaluation we partitioned each of these datasets into two, so that we could run two-fold crossvalidation: training on one half of the dataset and testing on the other half, and vice versa.

In addition, the British Library Sound Archive has a large collection of environmental sound recordings, and they made available to us a subset of 60 "dawn chorus" recordings. This consisted of 20 recordings each from three UK-based recordists, ranging in duration from 2 min to 20 min, and annotated by each recordist with a list of species heard (median 6 species per recording, range 3–12). We refer to this dataset as *bldawn*, and perform three-fold stratified crossvalidation: for each recordist, we train the system using the data from the other two recordists, and then test on the audio from the held-out recordist. This stratified approach is useful because it tests whether the system can generalise to recordings from unknown recordists, rather than adapting to any specifics of the known recordists.

We also gathered a single-label dataset as a subset of the recordings available from the Xeno Canto website[1], covering many of the common UK bird species, and covering at least all the species present in the *bldawn* dataset. We refer to this dataset as *xccoverbl*. For each species included, we queried Xeno Canto to retrieve three different recordings, preferring to retrieve recordings from the UK, but allowing the system to return recordings from further afield if too few UK recordings were available. Our search query also requested high-quality recordings (quality label 'A'), and song rather than calls, where possible. Since we retrieved three examples for each species, this enabled us to partition the dataset for three-fold crossvalidation: not stratified into individual recordists (as was *bldawn*), but sampled from a wide range of recordists.

---

[1] http://www.xeno-canto.org/

These datasets have widely varying characteristics, for example in the typical duration of the sound files, the recording location, and the number of classes to distinguish (Table 1). Note that most of the datasets have different and irreconcilable lists of class labels: in particular, for *bldawn* and *xccoverbl* the class label is the species, whereas *nips4b* and *lifeclef2014* use separate labels for song and calls. Of our datasets only *bldawn* and *xccoverbl* have strong overlap in their species lists. Therefore only these datasets could be combined to create larger pools of training data.

In this work we performed automatic classification for each audio file, without any segmentation procedure to select region(s) of bird vocalisation in the file. The only segmentation that is done is implicit in the collection processes for the dataset: for the two datasets originating from Xeno Canto, each audio clip might or might not contain a large amount of silence or other noise, depending on the contributor; for *nips4b* the audio is collected from remote monitoring stations with no manual selection; for *bldawn* the audio is selected by the contributor, but not trimmed to a specific vocalisation, instead selected to present a long dawn chorus audio recording.

## Feature learning method

As discussed in 'spectral features and feature learning', the aim of unsupervised feature learning is to find some transformation of a dataset, driven only by the characteristics inherent in that dataset. For this we use a method that has shown promise in previous studies, and can be run effectively at big data scales: *spherical k-means*, described by *Coates & Ng (2012)* and first applied to audio by *Dieleman & Schrauwen (2013)*. There are many feature-learning methods available, including neural networks such as restricted Boltzmann machines (used e.g., in *Erhan et al., 2010*), or methods based on sparse coding such as K-SVD (*Aharon, Elad & Bruckstein, 2006*). Our choice of method is motivated by the promising results of *Dieleman & Schrauwen (2013)* but also by our imperative to enable feature learning at very large scale. This leads to a preference for techniques of low computational complexity, and which can be applied to data in streaming fashion.

Spherical k-means is related to the simple and well-known k-means clustering algorithm (*Lloyd, 1982*), except that instead of searching for cluster centroids which minimise the Euclidean distance to the data points, we search for unit vectors (directions) to minimise their angular distance from the data points. This is achieved by modifying the iterative update procedure for the k-means algorithm: for an input data point, rather than finding the nearest centroid by Euclidean distance and then moving the centroid towards that data point, the nearest centroid is found by cosine distance,

$$\text{cosine distance} = 1 - \cos(\theta) = 1 - \frac{A \cdot B}{\|A\| \|B\|} , \tag{1}$$

where $A$ and $B$ are vectors to be compared, $\theta$ is the angle between them, and $\| \cdot \|$ is the Euclidean vector norm. The centroid is renormalised after update so that it is always a unit vector. Figure 1 shows an example of spherical k-means applied to synthetic data. Spherical k-means thus finds a set of unit vectors which represent the distribution of directions

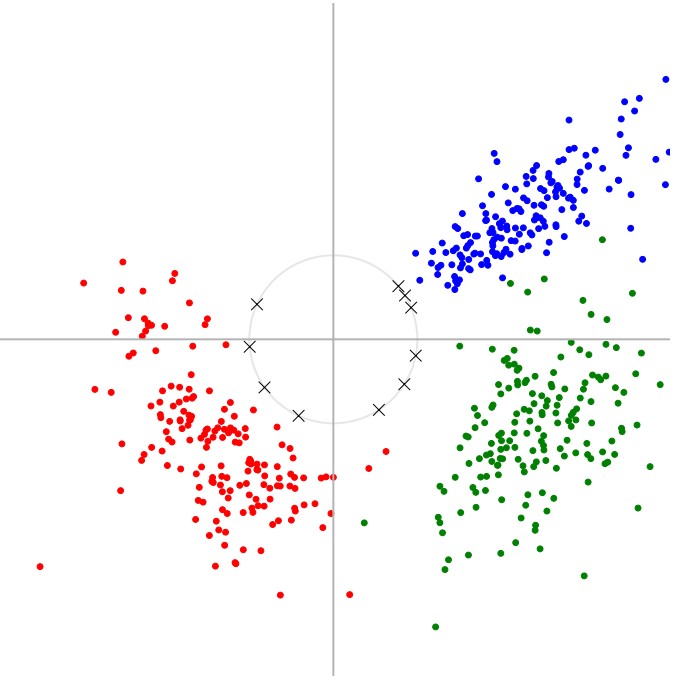

**Figure 1 Example of spherical k-means applied to a simple two-dimensional dataset.** We generated synthetic 2D data points by sampling from three clusters which were each Gaussian-distributed in terms of their angle and log-magnitude (coloured dots), and then applied our online spherical k-means algorithm to find 10 unit vectors (crosses). These unit vectors form an overcomplete basis with which one could represent this toy data, projecting two-dimensional space to ten-dimensional space.

found in the data: it finds a basis (here an overcomplete basis) so that data points can in general be well approximated as a scalar multiple of one of the basis vectors. This basis can then be used to represent input data in a new feature space which reflects the discovered regularities, in the simplest case by representing every input datum by its dot product with each of the basis vectors (*Coates & Ng, 2012*; *Dieleman & Schrauwen, 2013*):

$$x'(n,j) = \sum_{i=1}^{M} b_j(i) x(n,i), \tag{2}$$

where $x$ represents the input data indexed by time frame $n$ and feature index $i$ (with $M$ the number of input features, e.g., the number of spectral bins), $b_j$ is one of the learnt basis vectors (indexed by $j \in [1, k]$), and $x'$ is the new feature representation. In our case, the data on which we applied the spherical k-means procedure consisted of Mel spectral frames ($M = 40$ dimensions), which we first normalised and PCA-whitened as in *Dieleman & Schrauwen (2013)*.

We also tested configurations in which the input data was not one spectral frame but a sequence of them—e.g., a sequence of four spectral frames at a time—allowing the clustering to respond to short-term temporal patterns as well as spectral patterns. We can

write this as

$$x'(n,j) = \sum_{\delta=0}^{\Delta-1} \sum_{i=1}^{M} b_j(\delta,i) x(n+\delta,i) , \quad (3)$$

where $\Delta$ is the number of frames considered at a time, and the $b$ are now indexed by a frame-offset as well as the feature index. (See Fig. 10 to preview examples of such bases.) Alternatively, this can be thought of as "stacking" frames, e.g., stacking each sequence of four 40-dimensional spectral frames to give a 160-dimensional vector, before applying (2) as before. In all our experiments we used a fixed $k = 500$, a value which has been found useful in previous studies (*Dieleman & Schrauwen, 2013*).

The standard implementation of k-means clustering requires an iterative batch process which considers all data points in every step. This is not feasible for high data volumes. Some authors use "minibatch" updates, i.e., subsamples of the dataset. For scalability as well as for the potential to handle real-time streaming data, we instead adapted an online streaming k-means algorithm, "online Hartigan k-means" (*McFee, 2012*, Appendix B). This method takes one data point at a time, and applies a weighted update to a selected centroid dependent on the amount of updates that the centroid has received so far. We adapted the method of (*McFee, 2012*, Appendix B) for the case of spherical k-means. k-means is a local optimisation algorithm rather than global, and may be sensitive to the order of presentation of data. Therefore in order to minimise the effect of order of presentation for the experiments conducted here, we did not perform the learning in true single-pass streaming mode. Instead, we performed learning in two passes: a first streamed pass in which data points were randomly subsampled (using reservoir sampling) and then shuffled before applying PCA whitening and starting the k-means procedure, and then a second streamed pass in which k-means was further trained by exposing it to all data points. Our Python code implementation of online streaming spherical k-means is available as Supplemental Information.

As a further extension to the method, we also tested a *two-layer* version of our feature-learning method, intended to reflect detail across multiple temporal scales. In this variant, we applied spherical k-means feature learning to a dataset, and then projected the dataset into that learnt space. We then downsampled this projected data by a factor of 8 on the temporal scale (by max-pooling, i.e., taking the max across each series of 8 frames), and applied spherical k-means a second time. The downsampling operation means that the second layer has the potential to learn regularities that emerge across a slightly longer temporal scale. The two-layer process overall has analogies to deep learning techniques, most often considered in the context of artificial neural networks (*Erhan et al., 2010*; *Bengio, Courville & Vincent, 2013*), and to the progressive abstraction believed to occur towards the higher stages of auditory neural pathways.

## Classification and evaluation

Our full classification workflow started by converting each audio file to a standard sample-rate of 44.1 kHz. We then calculated Mel spectrograms for each file, using a frame

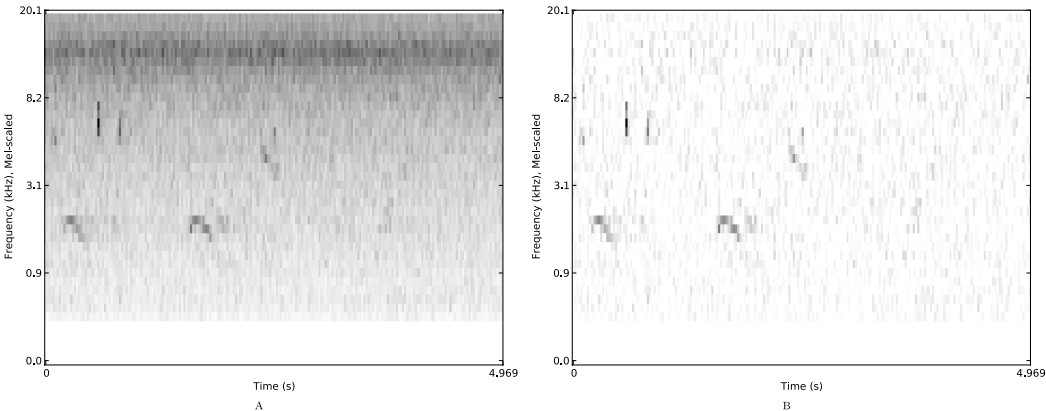

**Figure 2  Mel spectrograms of a single example from the *nips4b* dataset.** Mel spectrograms of a single example from the *nips4b* dataset, with median-based noise reduction off (A) or on (B).

size of 1024 frames with Hamming windowing and no overlap. We chose no overlap rather than 50% overlapped frames simply to reduce the volume of data to be processed. We filtered out spectral energy below 500 Hz, a heuristic choice which strongly reduces the amount of environmental noise present, a benefit which is traded off against the cost that this will discard some energy from species that vocalise below 500 Hz (such as the eagle-owl *Bubo bubo*). We then normalised the root-mean-square (RMS) energy in each spectrogram.

For each spectrogram we then optionally applied the noise-reduction procedure that we had found to be useful in our NIPS4B contest submission (*Stowell & Plumbley, 2013*), a simple and common median-based thresholding. This consists of finding the median value for each spectral band in a spectrogram, then subtracting this median spectrum from every frame, and setting any resulting negative values to zero. This therefore preserves only the spectral energy that rises above the median bandwise energy. In principle it is a good way to reduce the stationary noise background (Fig. 2), but is not designed to cope well with fluctuating noise. However its simplicity makes it easy to apply across large datasets efficiently.

The Mel spectrograms, either noise-reduced or otherwise, could be used directly as features. We also tested their reduction to MFCCs (including delta features, making 26-dimensional data), and their projection onto learned features, using the spherical k-means method described above. For the latter option, we tested projections based on single frame as well as on sequences of 2, 3, 4 and 8 frames, to explore the benefit of modelling short-term temporal variation. We also tested the two-layer version based on the repeated application to 4-frame sequences across two timescales.

The feature representations thus derived were all time series. In order to reduce them to summary features for use in the classifier, we tested two common and simple techniques: summarising each feature dimension independently by its mean and standard deviation, or alternatively by its maximum. These are widespread but are not designed to reflect any temporal structure in the features (beyond the fine-scale temporal information

**Table 2  The twelve combinations of feature-type and feature-summarisation tested.** The feature-type and feature-summarisation method jointly determine the dimensionality of the data input to the classifier.

| Label | Features | Summarisation | Dimension |
|---|---|---|---|
| `mfcc-ms` | MFCCs (+deltas) | Mean & stdev | 52 |
| `mfcc-maxp` | MFCCs (+deltas) | Max | 26 |
| `mfcc-modul` | MFCCs (+deltas) | Modulation coeffs | 260 |
| `melspec-ms` | Mel spectra | Mean & stdev | 80 |
| `melspec-maxp` | Mel spectra | Max | 40 |
| `melspec-modul` | Mel spectra | Modulation coeffs | 400 |
| `melspec-kfl1-ms` | Learned features, 1 frame | Mean & stdev | 1,000 |
| `melspec-kfl2-ms` | Learned features, 2 frames | Mean & stdev | 1,000 |
| `melspec-kfl3-ms` | Learned features, 3 frames | Mean & stdev | 1,000 |
| `melspec-kfl4-ms` | Learned features, 4 frames | Mean & stdev | 1,000 |
| `melspec-kfl8-ms` | Learned features, 8 frames | Mean & stdev | 1,000 |
| `melspec-kfl4pl8kfl4-ms` | Learned features, 4 frames, two-layer | Mean & stdev | 1,000 |

that is captured by some of our features). Therefore, for the Mel and MFCC features we also tested summarising by modulation coefficients: we took the short-time Fourier transform (STFT) along the time axis of our features, and then downsampled the spectrum to a size of 10 to give a compact representation of the temporal evolution of the features (cf. *Lee, Han & Chuang, 2008*). The multi-frame feature representations already intrinsically included short-term summarisation of temporal variation, so to limit the overall size of the experiment, for the learned feature representations we only applied the mean-and-standard-deviation summarisation. Overall we tested six types of non-learned representation against six types of learned representation (Table 2).

To perform classification on our temporally-pooled feature data, then, we used a random forest classifier (*Breiman, 2001*). A random forest classifier is an ensemble method which trains many decision-tree classifiers on the same dataset: the decision trees are different from each other due to the use of "bagging"—drawing a different bootstrap sample from the training dataset for each tree—and also by considering only a small random subset of the available data features as candidates for each split. This randomisation reduces the correlation between individual decision trees. To make a prediction, the random forest uses a simple vote to aggregate the predictions of its decision trees: in this work we use probabilistic outputs from the classifier, meaning that the vote proportions are reinterpreted as probabilities. Random forests and other tree-ensemble classifiers perform very strongly in a wide range of empirical evaluations (*Caruana & Niculescu-Mizil, 2006*), and were used by many of the strongest-performing entries to the SABIOD evaluation contests (*Glotin et al., 2013*; *Fodor, 2013*; *Potamitis, 2014*). For this experiment we used the implementation from the Python `scikit-learn` project (*Pedregosa et al., 2011*). Note that `scikit-learn` v0.14 was found to have a specific

issue preventing training on large data, so we used a pre-release v0.15 after verifying that it led to the same results with our smaller datasets.

We did not manually tune any parameters of the classifier: parameter tuning can lead to improvements in performance, but can also lead to overfitting to particular dataset characteristics, so in all cases we trained a random forest with 200 trees using the 'entropy' (information-gain) criterion to measure the quality of a split. However, since our experiment covered both single-label and multilabel classification, we did test three different ways of using the classifier to make decisions:

1. Single-label classification: this assumes that there is only one species present in a recording. It therefore cannot be applied to multilabel datasets, but for single-label datasets it may benefit from being well-matched to the task.
2. Binary relevance: this divides the multilabel classification task into many single-label tasks, training one separate classifier for each of the potential output labels (*Tsoumakas, Katakis & Vlahavas, 2010*). This strategy ignores potential correlations between label occurrence, but potentially allows a difficult task to be approximated as the combination of more manageable tasks. Binary relevance is used e.g., by *Fodor (2013)*.
3. Full multilabel classification: in this approach, a single classifier (here, a single random forest) is trained to make predictions for the full multi-label situation. Predicting presence/absence of every label simultaneously can be computationally difficult compared against a single-label task, and may require larger training data volumes, but represents the full situation in one model (*Tsoumakas, Katakis & Vlahavas, 2010*).

For each of these methods the outputs from the classifier are per-species probabilities. We tested all of our datasets using the full multilabel classifier, then for comparison we tested the single-label datasets using the single-label classifier, and the multi-label datasets using the binary-relevance classifier.

Some of our datasets contain long audio recordings, yet none of the annotations indicate which point(s) in time each species is heard. This is a common format for annotations: for example, the *bldawn* annotations are derived directly from the archival metadata, which was not designed specifically for automatic classification. Long audio files present an opportunity to make decisions either for the entire file as one scene, or in smaller "decision windows", for which the decisions are then pooled to yield overall decisions. We tested this empirically, using decision windows of length 1, 5 or 60 s or the whole audio. Each decision window was treated as a separate datum for the purposes of training and testing the classifier, and then the decisions were aggregated per audio file using either mean or maximum. The mean probability of a species across all the decision windows is a reasonable default combination; we compared this against the maximum with the motivation that if a bird is heard only at one point in the audio, and this leads to a strong detection in one particular decision window, then such a strong detection should be the overriding factor in the overall decision. For some datasets (*nips4b*) we did not test long windows since all audio files were short; while for *lifeclef2014* we used only

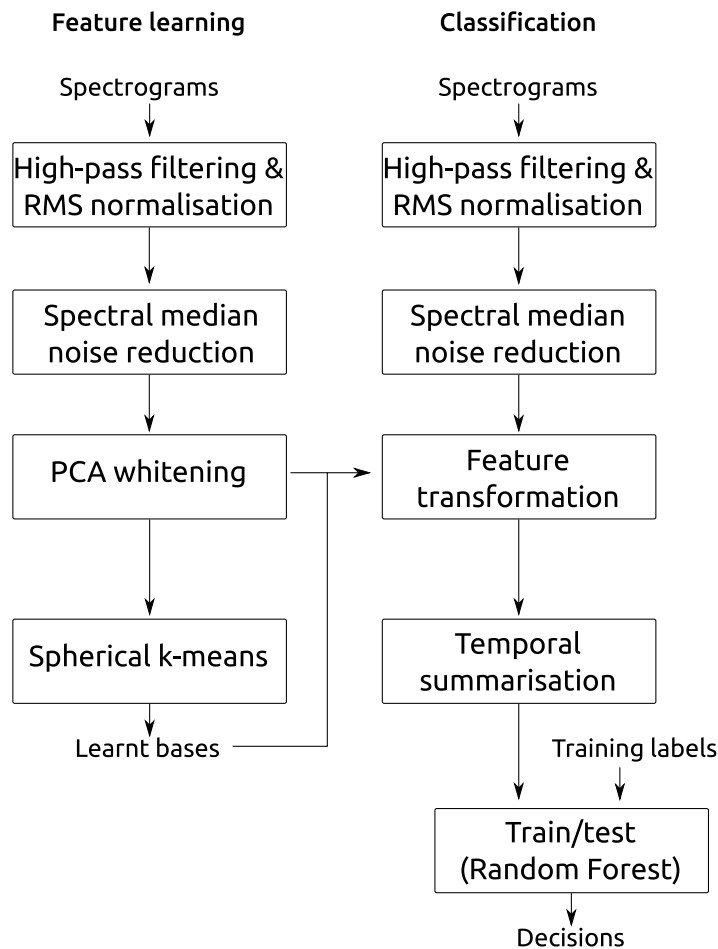

**Feature learning**   **Classification**

**Figure 3** **Summary of the classification workflow.** Summary of the classification workflow, here showing the case where single-layer feature learning is used.

whole-audio classification because of the runtime costs of evaluating these combinations over this largest dataset.

We performed feature learning, training and testing separately for each of our four datasets, using the appropriate two- or threefold crossvalidation described above, and across all combinations of the feature settings we have just described. Figure 3 summarises the main stages of the workflow described.

We evaluated the performance in each experimental run using two measures, the Area Under the ROC Curve (AUC) and the Mean Average Precision (MAP). The AUC statistic is an evaluation measure for classification/detection systems which has many desirable properties (*Fawcett, 2006*): unlike raw accuracy, it is not affected by "unbalanced" datasets having an uneven mixture of true-positive and true-negative examples; and it has a standard probabilistic interpretation, in that the AUC statistic tells us the probability that the algorithm will rank a random positive instance higher than a random negative instance. Chance performance is always 50% for the AUC statistic. The AUC is a good statistic to use to evaluate the output from a probabilistic classifier in general, but for

**Peer**J

user-facing applications, which may for example show a ranked list of possible hits, the Mean Average Precision (MAP) statistic leads to an evaluation which relates more closely to user satisfaction with the ranked list. The key difference in effect is that the AUC statistic applies an equal penalty for a misordering at any position on the ranked list, whereas the MAP statistic assigns greater penalties higher up the ranking (*Yue et al., 2007*). We therefore calculate both evaluation statistics.

To test for significant differences in the performance statistics, we applied a generalised linear model (GLM) using the `lme4` package (*Bates et al., 2014*) for R 2.15.2 (*R Core Team, 2012*). We focused primarily on the AUC for significance testing, since the AUC and MAP statistics are related analyses of the same data. Since AUC is bounded in the range $[0, 1]$, we applied the GLM in the logistic domain: note that given the probabilistic interpretation of AUC, the logistic model is equivalent to applying a linear GLM to odds ratios, a fact which facilitates interpretation. Every experimental run for a given dataset used the same set of folds, so we used a repeated-measures version of the GLM with the "fold index" as the grouping variable. We tested for individual and pairwise interactions of our five independent categorical variables, which were as follows:

- choice of feature set and temporal summarisation method, testing the 12 configurations listed in Table 2;
- noise reduction on vs. off;
- classifier mode (multilabel vs. either single-label or binary-relevance);
- decision pooling window duration (1, 5, or 60 s or whole audio);
- decision pooling max vs. mean.

Combinatorial testing of all these configurations resulted in $12 \times 2 \times 2 \times 4 \times 2 = 384$ crossvalidated classification experiments for the *bldawn* and *xccoverbl* datasets. For the other datasets we did not test all four pooling durations, for reasons given above: the number of crossvalidated experiments was thus 192 for *nips4b* and 96 for *lifeclef2014*. Since the tests of *lifeclef2014* did not vary decision pooling, decision pooling factors were not included in that GLM. We considered effects to be significant when the 95% confidence interval calculated from the GLM excluded zero, in which cases we report the estimated effects as differences in odds-ratios.

## Additional tests

The *bldawn* dataset has relatively few annotations, since it only consists of 60 items. We therefore wanted to explore the use of auxiliary information from other sources to help improve recognition quality, in particular using the *xccoverbl* dataset, which has strong overlap in the list of species considered. In further tests we tested three ways of using this additional data:

1. **Cross-condition training**, meaning training on one dataset and testing on the other. The two datasets have systematic differences—for example, *xccoverbl* items are annotated with only one species each, and are generally shorter—and so we did not expect this to yield very strong results.

2. **Data augmentation** for the feature learning step, meaning that feature learning is conducted using the training data for the *bldawn* as well as all of the *xccoverbl* data. This gives a larger and more varied pool of data for the feature learning step, which we expected to give a slight improvement to the results of feature learning.

3. **Data augmentation** for feature learning and also for training. Although the systematic differences mean the *xccoverbl* training data might not guide the classifier in the correct way, it holds many more species annotations for the species of interest, in a wider set of conditions. We expected the combined training would provide stronger generalisation performance.

We evaluated these train/test conditions as separate evaluation runs.

We also wanted to distinguish between two possible explanations for any difference between the performance of the different feature sets: was it due to intrinsic characteristics of the features, or more simply due to the dramatic differences in feature dimensionality (which ranged 26–1,000; see Table 2)? Differences in dimensionality might potentially give more degrees of freedom to the classifier without necessarily capturing information in a useful way. In order to test this, we ran a version of our test in which for each experimental run we created a random projection matrix which projected the feature set to a fixed dimensionality of 200. For MFCC/Mel features this was a simple form of data expansion, while for learned features it was a form of data reduction. By standardising the feature dimensionality, this procedure decoupled the nature of the feature set from the degrees of freedom available to the classifier. We ran this test using the *nips4b* dataset.

## RESULTS

Recognition performance was generally strong (Figs. 4 and 5, Table 3 ), given the very large number of classes to distinguish (at least 77). The AUC and MAP performance measures both led to very similar rankings in our experiments.

The strongest effect found in our tests was the effect of feature type, with a broad tendency for MFCCs to be outperformed by Mel spectra, and both of these outperformed by learned features. For the largest dataset, *lifeclef2014*, feature learning led to classification performance up to 85.4% AUC, whereas without feature learning the performance peaked at 82.2% for raw Mel spectra or 69.3% for MFCCs. This pattern was clear for all datasets except *bldawn*. Compared against the baseline standard configuration `mfcc-ms`, switching to learned features provided all the strongest observed boosts in recognition performance (Table 3). The effect was particularly strong for the two single-label datasets, *xccoverbl* and *lifeclef2014* (effect size estimates $\geq 0.86$ for all feature-learning variants). For *nips4b* there was a milder effect ($\approx 0.25$), except for the two-layer version which had a significant negative effect ($-0.36$). Conversely, the two-layer version achieved strongest performance on the largest dataset (*lifeclef2014*). These facts together suggest that the performance impairment for *nips4b* was due to the relatively small size of the dataset, since deeper models typically require more data (*Coates & Ng, 2012*). That aside, the performance differences between variants of our feature-learning method were small. For all datasets except *bldawn*, the switch from MFCCs to raw Mel spectral features also provided a

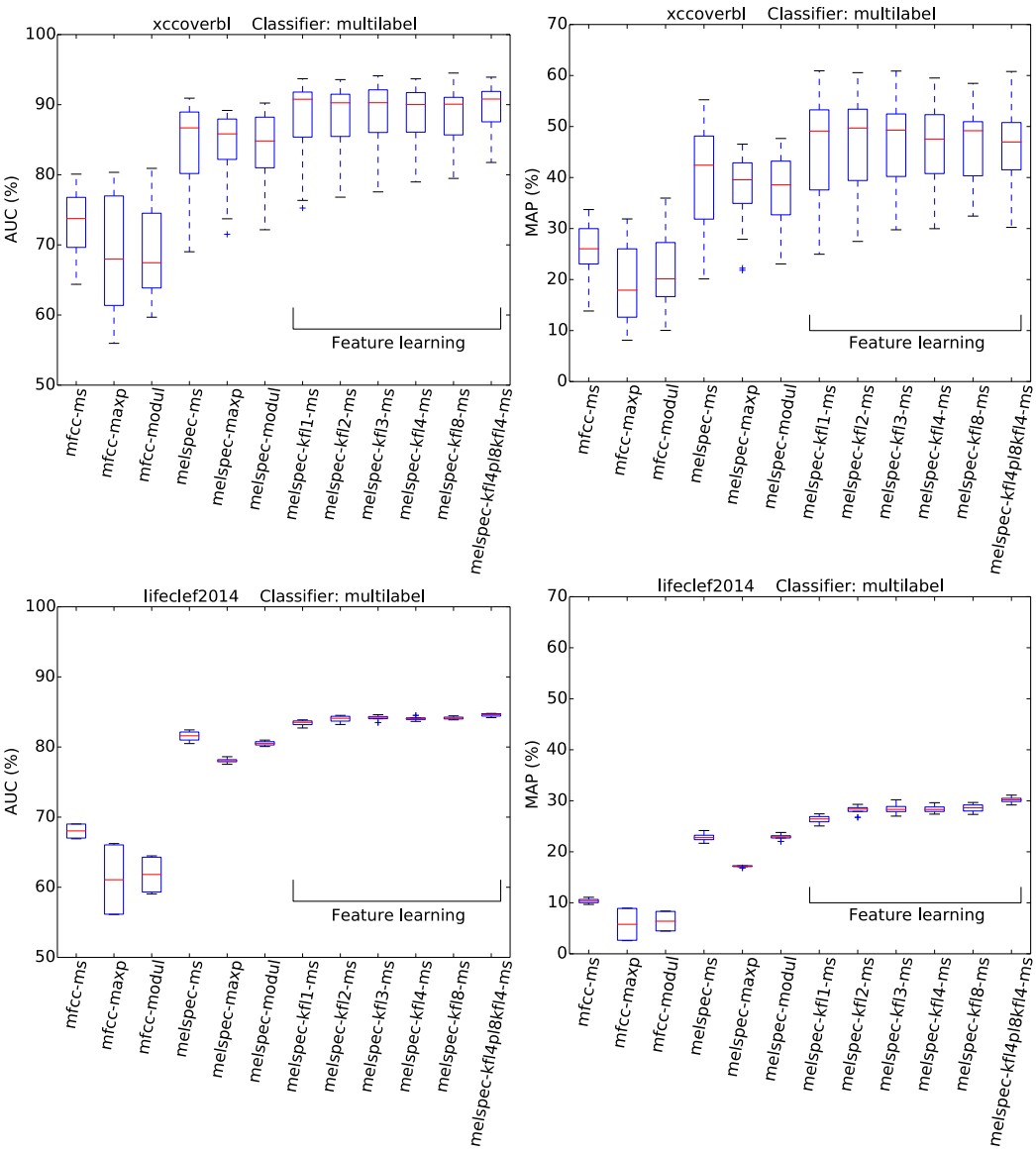

**Figure 4 AUC and MAP statistics, summarised for each feature-type tested—here for the two single-label datasets, using the full multilabel classifier.** Boxes span the quartiles of the values attained, while the whiskers indicate the full range. Each column in the boxplot summarises the crossvalidated scores attained over many combinations of the other configuration settings tested (for the full multi-class classifier only). The ranges indicated therefore do not represent random variation due to training data subset, but systematic variation due to classifier configuration. Figure 5 plots the same for the multilabel datasets.

strong boost in performance, though not to the same extent as did the learned features. Across those three datasets, mean-and-standard-deviation summarisation consistently gave the strongest performance over our two alternatives (i.e., maximum or modulation coefficients).

None of the above tendencies are discernible in the results for *bldawn*, for which all methods attain the same performance. The classifier can reach over 80% AUC (50% MAP),

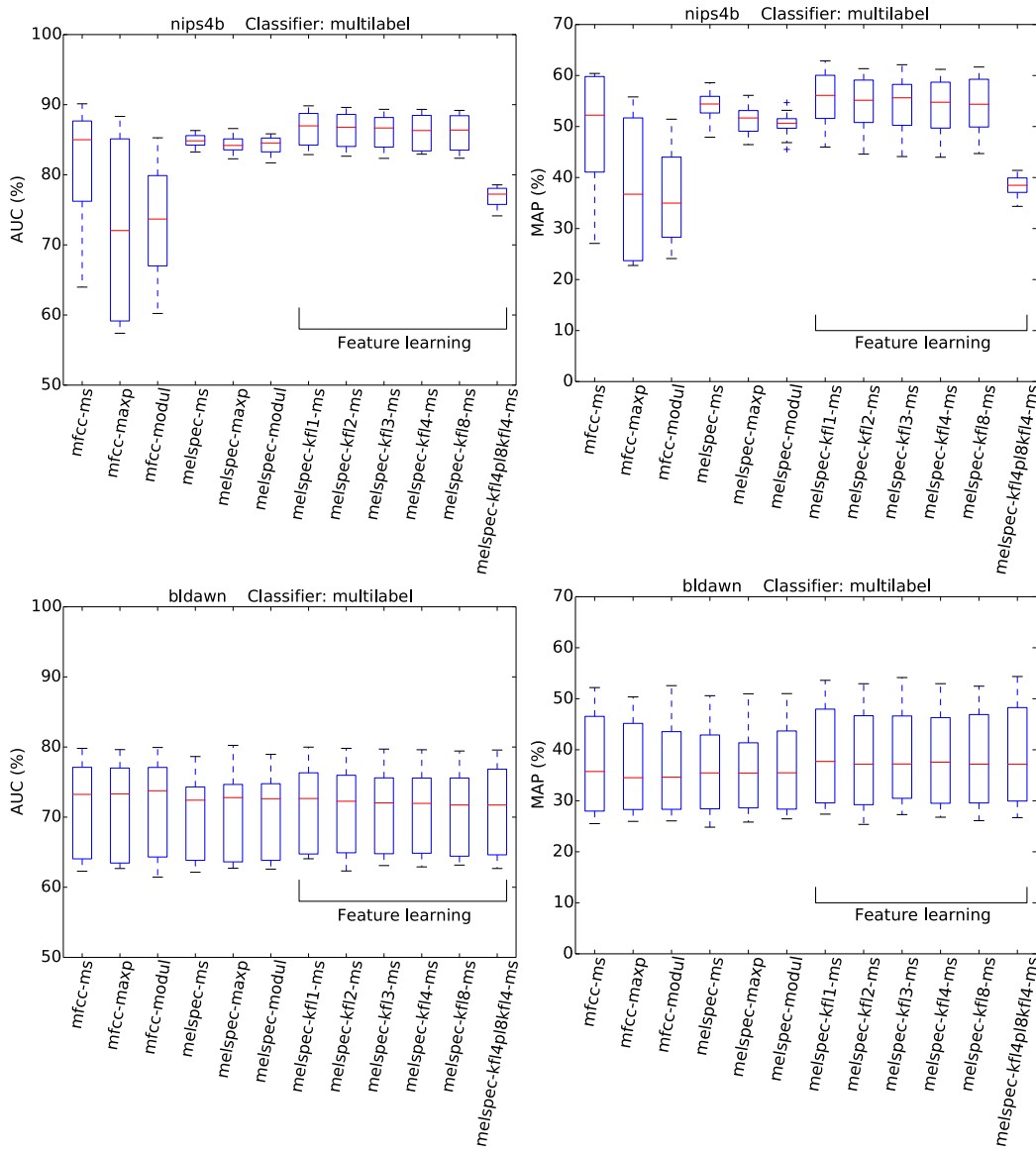

**Figure 5** AUC and MAP statistics, summarised for each feature-type tested—here for the two multilabel datasets, with the multilabel classifier.

which is far above chance performance, but not as strong as for the other datasets, nor showing the pattern of differentiation between the configurations. The relatively low scores reflect a relatively low ability to generalise, demonstrated by the observation that the trained systems attained very strong scores when tested on their training data (>99.95% in all cases, and 100% in most). This outcome is typical of systems trained with an amount of ground-truth annotations which is insufficient to represent the full range of variation within the classes.

The choice of classifier mode showed moderate but consistent effects across all the combinations tested. For multilabel datasets, a decrease in AUC was observed by switching

**Table 3** **First-order effect sizes, estimated by our GLM as linear changes to the AUC odds ratio.** Significant results are marked in bold with an asterisk, judged relative to a baseline category indicated in the first column. Positive values represent an improvement over the baseline. Empty cells indicate combinations that were not tested, as described in the text.

| Factor | Factor value | nips4b | xccoverbl | bldawn | lifeclef2014 |
|---|---|---|---|---|---|
| featureset (vs. mfcc-ms) | mfcc-maxp | * −0.59 | * −0.29 | −0.03 | * −0.30 |
| | mfcc-modul | * −0.55 | * −0.45 | * −0.12 | * −0.27 |
| | melspec-ms | * 0.10 | * 1.01 | * −0.04 | * 0.73 |
| | melspec-maxp | −0.01 | * 0.82 | −0.03 | * 0.52 |
| | melspec-modul | 0.02 | * 0.82 | −0.03 | * 0.67 |
| | melspec-kfl1-ms | * 0.26 | * 1.43 | 0.01 | * 0.86 |
| | melspec-kfl2-ms | * 0.25 | * 1.36 | −0.02 | * 0.90 |
| | melspec-kfl3-ms | * 0.23 | * 1.44 | −0.00 | * 0.92 |
| | melspec-kfl4-ms | * 0.20 | * 1.40 | −0.00 | * 0.91 |
| | melspec-kfl8-ms | * 0.21 | * 1.39 | −0.01 | * 0.91 |
| | melspec-kfl4pl8kfl4-ms | * −0.36 | * 1.40 | −0.00 | * 0.95 |
| noisered. | on | * −0.64 | * −0.20 | −0.01 | * 0.06 |
| pooldur (vs. none) | 1 | * −0.23 | * −0.15 | −0.02 | |
| | 5 | | * −0.15 | −0.04 | |
| | 60 | | | 0.04 | −0.00 |
| dpoolmode | mean | 0.03 | * 0.07 | 0.01 | |
| classif (vs. multi) | binary relevance | * −0.28 | | * −0.05 | |
| | single-label | | 0.05 | | 0.01 |

to the "binary relevance" approach to classification. Note however that this difference is more pronounced for AUC than for MAP (Fig. 6). For single-label datasets, no significant effect was observed, with a very small boost in AUC by switching from the multilabel classifier to the single-label classifier.

Splitting the audio into decision windows and then combining the outcomes generally had a negative or negligible effect on outcomes; however, using mean (rather than maximum) to aggregate such decisions had a mild positive effect (significant only for *xccoverbl*). Looking at the second-order interaction did not find any synergistic positive effects of using mean-pooling and a particular window length. Activating noise reduction showed an inconsistent effect, significantly impairing performance on *nips4b* (and to a lesser extent *xccoverbl*) while slightly improving performance on *lifeclef2014*.

Our follow-up data expansion tests and cross-condition test failed to improve performance for *bldawn* (Fig. 7). Adding the *xccoverbl* data to the feature learning step made little difference, giving a slight but insignificant boost to the two-layer model. This tells us firstly that the dataset already contained enough audio for feature-learning to operate satisfactorily, and secondly that the audio from *xccoverbl* is similar enough in kind that its use is no detriment. However, the story is quite different for the cases in which we then included *xccoverbl* in the training step. With or without feature learning, using *xccoverbl* to provide additional training data for *bldawn* acted as a distractor in this particular classification setup, and performed uniformly poorly. Note that the data

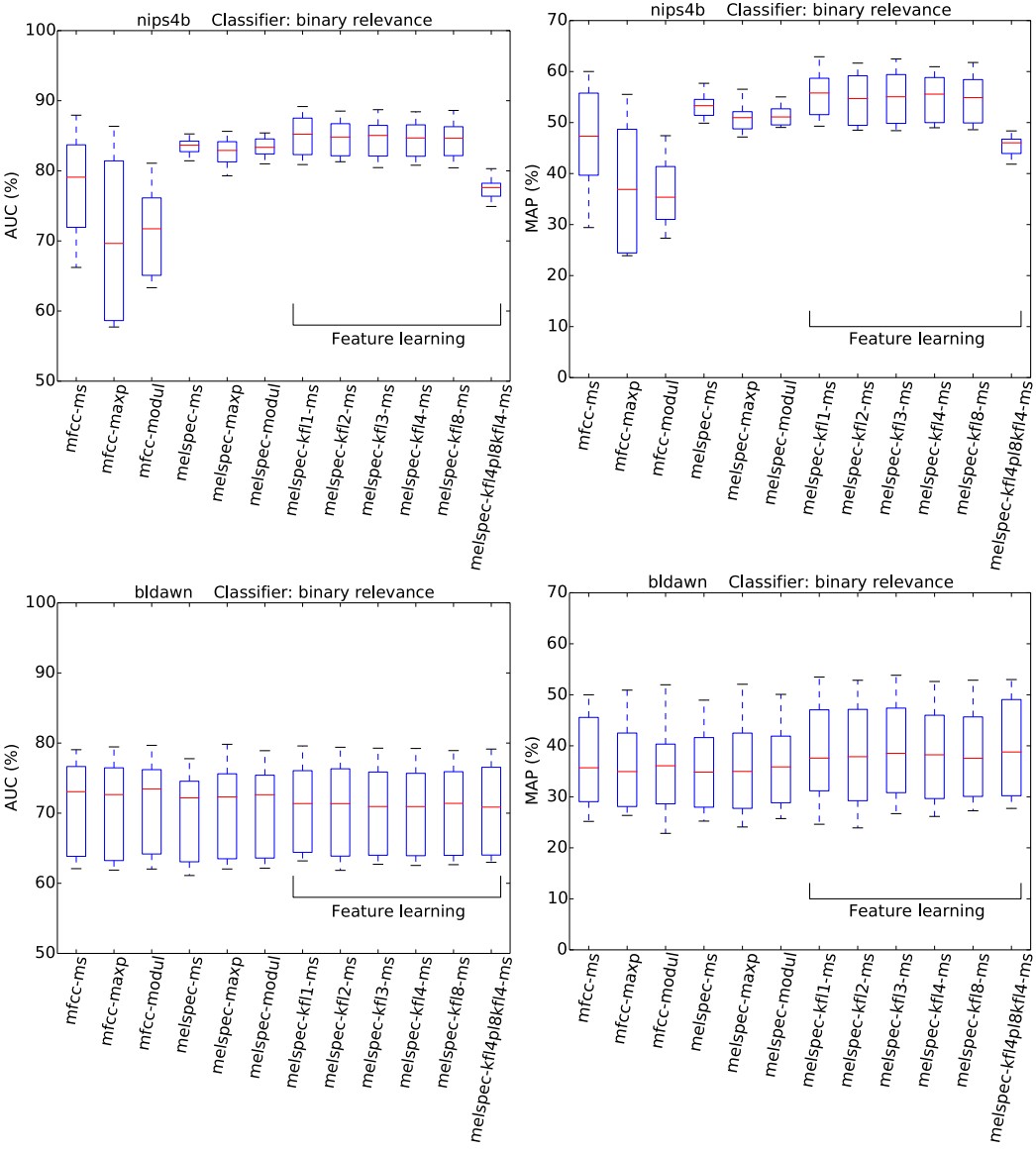

**Figure 6  AUC and MAP statistics, summarised for each feature-type tested—here for the two multil-abel datasets, with the binary relevance classifier.**

expansion procedure augmented the audio data size by around 60%, but augmented the number of annotations even more substantially (since *xccoverbl* contains more individual items than *bldawn*), and so the classifier may have been led to accommodate the single-label data better than the dawn chorus annotations. We diagnosed the problem by looking at the classification quality that the classifiers attained on their *bldawn* training examples: in this case the quality was poor (AUC < 60%), confirming that the single-label *xccoverbl* data had acted as distractors rather than additional educational examples.

Turning to the *nips4b* dataset to explore the effect of feature dimensionality, the relative performance of the different feature types was broadly preserved even after

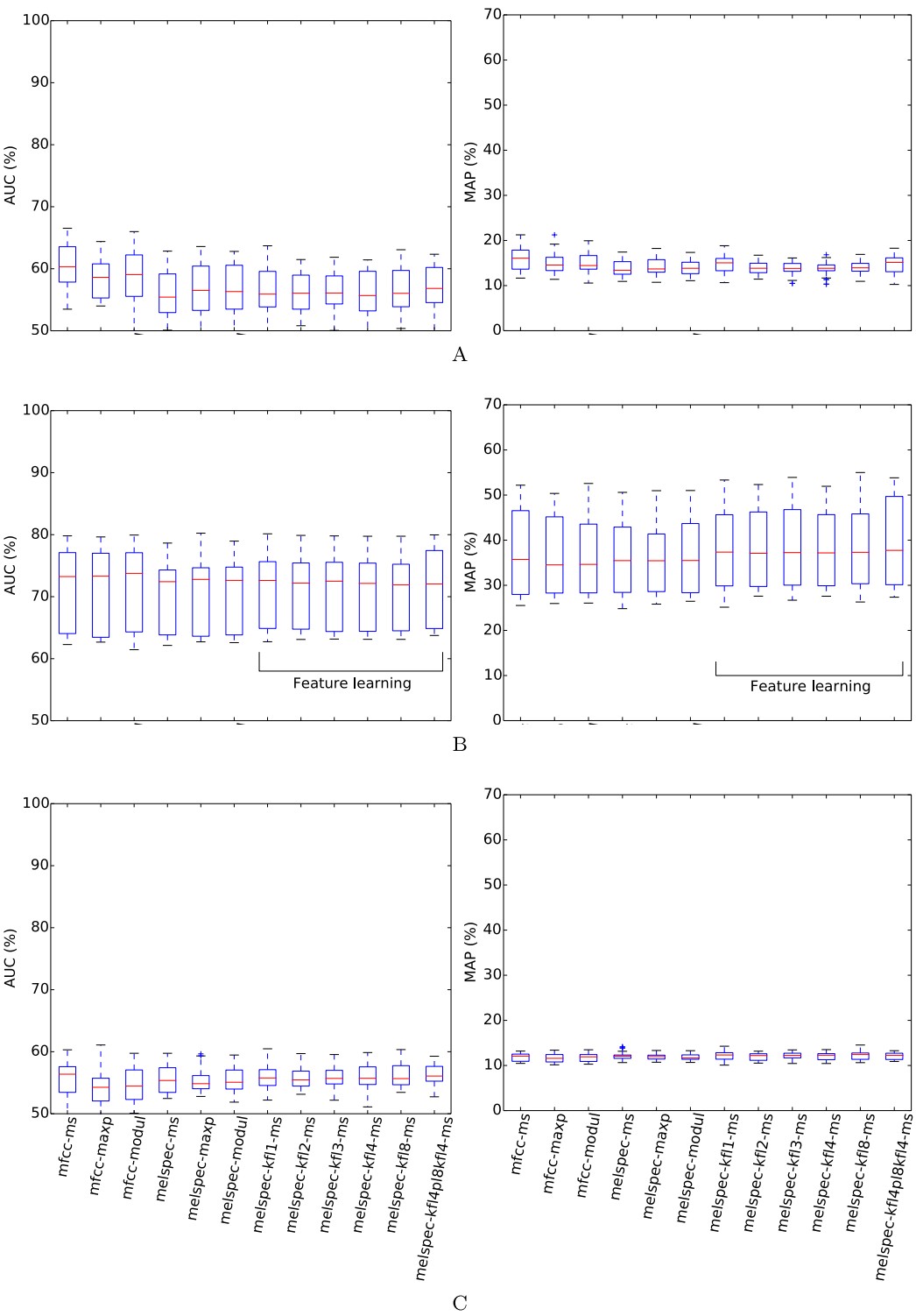

**Figure 7 AUC and MAP statistics, summarised for each feature-type tested—here for the *bldawn* dataset, but testing three different ways of making use of the *xccoverbl* data.**

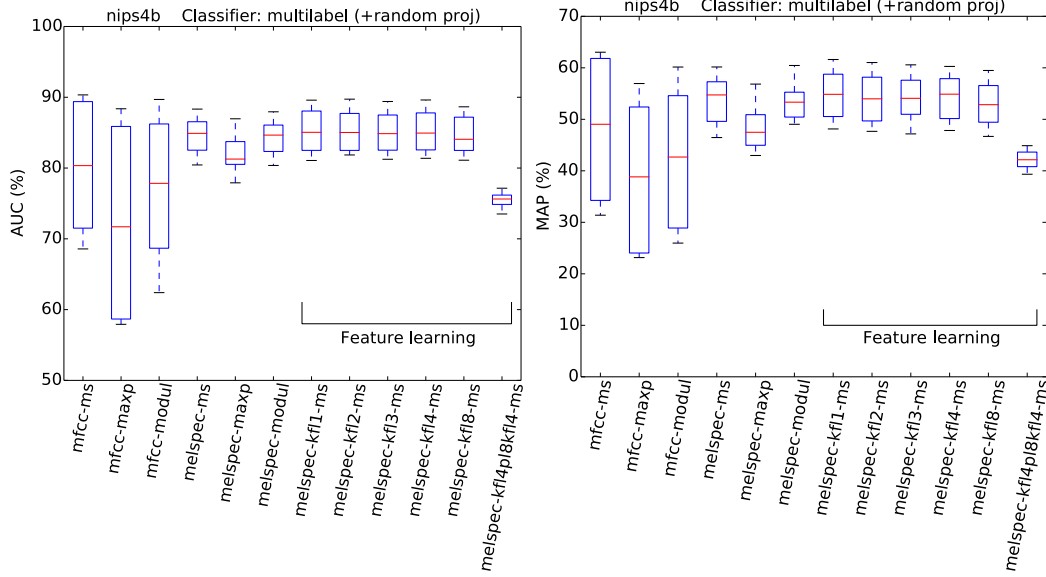

**Figure 8 AUC and MAP statistics, summarised for each feature-type tested—here for the *nips4b* dataset, but also using random projection.** In this variant of the experiment, the feature dimensionality is standardised before train/test by the application of a random projection.

dimensionality was standardised by random projection (Fig. 8). The general effect of the random projection was a modest improvement for low-dimensional features, and a modest impairment for high-dimensional (learned) features, but not to the extent of changing the ordering of performance. This suggests that high dimensionality is a small, though non-zero, part of what lends the learned features their power. The overall effect estimated by the GLM for the random-projection modification was a small but significant impairment ($-0.07$).

We can compare our results against those obtained recently by others. A formal comparison was conducted in Spring 2014 when we submitted decisions from our system to the LifeCLEF 2014 bird identification challenge (*Goëau et al., 2014*). In that evaluation, our system attained by far the strongest audio-only classification results, with a MAP peaking at 42.9% (Table 4). (Only one system outperformed ours, peaking at 51.1% in a variant of the challenge which provided additional metadata as well as audio.) We submitted the outputs from individual models, as well as model-averaging runs using the simple mean of outputs from multiple models. Notably, the strongest classification both in our own tests and the official evaluation was attained not by model averaging, but by a single model based on two-layer feature learning. Also notable is that our official scores, which were trained and tested on larger data subsets, were substantially higher than our crossvalidated scores, corroborating our observation that the method works particularly well at high data volumes.

Considering the *nips4b* dataset, the peak result from our main tests reached a crossvalidated AUC of 89.8%. In the actual NIPS4B contest (conducted before our current approach was developed), the winning result attained 91.8% (*Glotin et al., 2013*);

**Table 4** **Summary of MAP scores attained by our system in the public LifeCLEF 2014 Bird Identification Task** (*Goëau et al., 2014*). The first column lists scores attained locally in our two-fold *lifeclef2014* split. The second column lists scores evaluated officially, using a classifier(s) trained across the entire training set.

| System variant submitted | Cross-validated MAP (%) | Final official MAP (%) |
|---|---|---|
| `melspec-kfl3-ms`, noise red., binary relevance | 30.56 | 36.9 |
| Average from 12 single-layer models | 32.73 | 38.9 |
| `melspec-kfl4pl8kfl4-ms`, noise red., binary relevance | **35.31** | **42.9** |
| Average from 16 single- and double-layer models | 35.07 | 41.4 |

*Potamitis (2014)*, developing further a model submitted to the contest, reports a peak AUC of 91.7%. Our results are thus slightly behind these, although note that these other reported results use the full public-and-private datasets, without crossvalidation, whereas we restricted ourselves only to the data that were fully public and divided this public data into two crossvalidation folds, so the comparison is not strict.

We measured the total time taken for each step in our workflow, to determine the approximate computational load for the steps (Fig. 9). The timings are approximate—in particular because our code was modularised to save/load state on disk between each process, which impacted particularly on the "classify" step which loaded large random forest settings from disk before processing. Single-layer feature learning was efficient, taking a similar amount of time as did the initial feature extraction. Double-layer feature learning took more than double this, because of the two layers as well as performing max-pooling downsampling. Training the random forest classifier took longer on the learned features due to the higher dimensionality. However, once the system was trained, the time taken to classify new data was the same across all configurations.

## DISCUSSION

In order to be of use for applications in ecology and archival, automatic bird species recognition from sound must work across large data volumes, across large numbers of potential species, and on data with a realistic level of noise and variation. Our experiments have demonstrated that very strong results can be achieved in exactly these cases by supplementing a classification workflow with unsupervised feature learning. We have here used a random forest classifier, but unsupervised feature learning operates without any knowledge of the classifier or even the training labels, so we can expect this finding to apply in other classification systems (cf. *Erhan et al., 2010*). The procedure requires large data volumes in order for benefits to be apparent, as indicated by the failure of two-layer feature learning on the *nips4b* dataset. However, the use of single-layer feature learning creates a classifier that is equivalent to or better than manually-designed features in all our tests. There were very few differences in performance between our different versions of feature learning. One difference is that two-layer feature learning, while unsuccessful for *nips4b*, led to the strongest performance for *lifeclef2014* which is the largest dataset

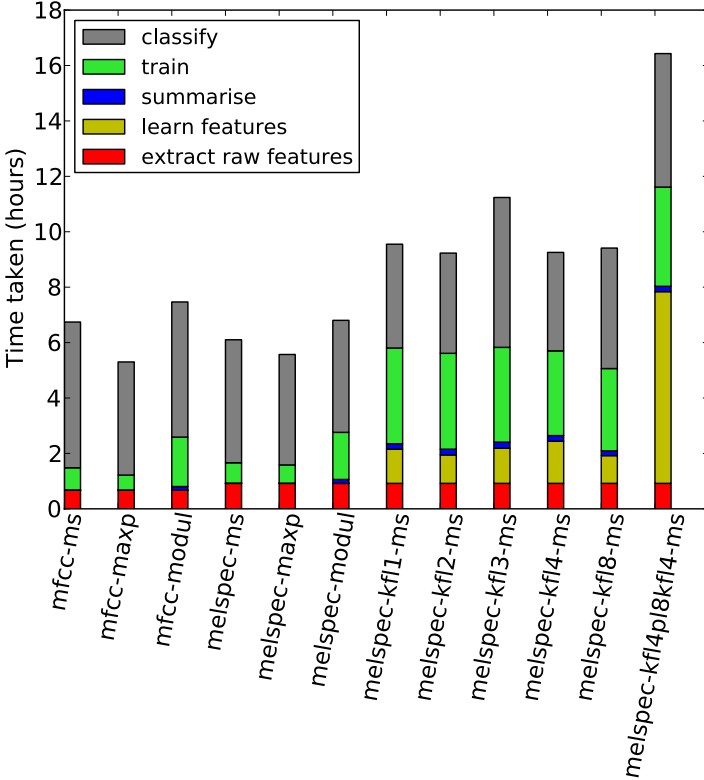

**Figure 9 Times taken for each step in the process, for the *lifeclef2014* dataset.** Note that these are heuristic "wallclock" times measured on processes across two compute servers, and disk read/write processes (to store state) took non-trivial time in each step. Each measurement is averaged across the two folds and across two settings (noise reduction on/off) across the runs using the multilabel classifier and no decision-pooling.

considered—largest by an order of magnitude in data volume, and by almost an order of magnitude in the number of possible species labels. This confirms the recommendations of *Coates & Ng (2012)* about the synergy between feature learning and big data scales, here for the case of ecological audio data.

However, note that a lesser but still substantial improvement over the baseline MFCC system can usually be attained simply by using the raw Mel spectral data as input rather than MFCCs. One of the long-standing motivations for the MFCC transformation has been to reduce spectral data down to a lower dimensionality while hoping to preserve most of the implicit semantic information; but as we have seen, the random forest classifier performs well with high-dimensional input, and such data reduction is not necessary and often holds back classification performance. Future investigators should consider using Mel spectra as a baseline, perhaps as an alternative to MFCCs as is common at present.

The lack of improvement on the *bldawn* dataset is notable, along with the low-quality results obtained by simply using a different dataset for training, or augmenting the data with an additional dataset. The availability of audio data is not the issue here, as the dataset is second-largest by audio volume, and augmenting the feature-learning step with additional data made little difference. The availability of training annotations may

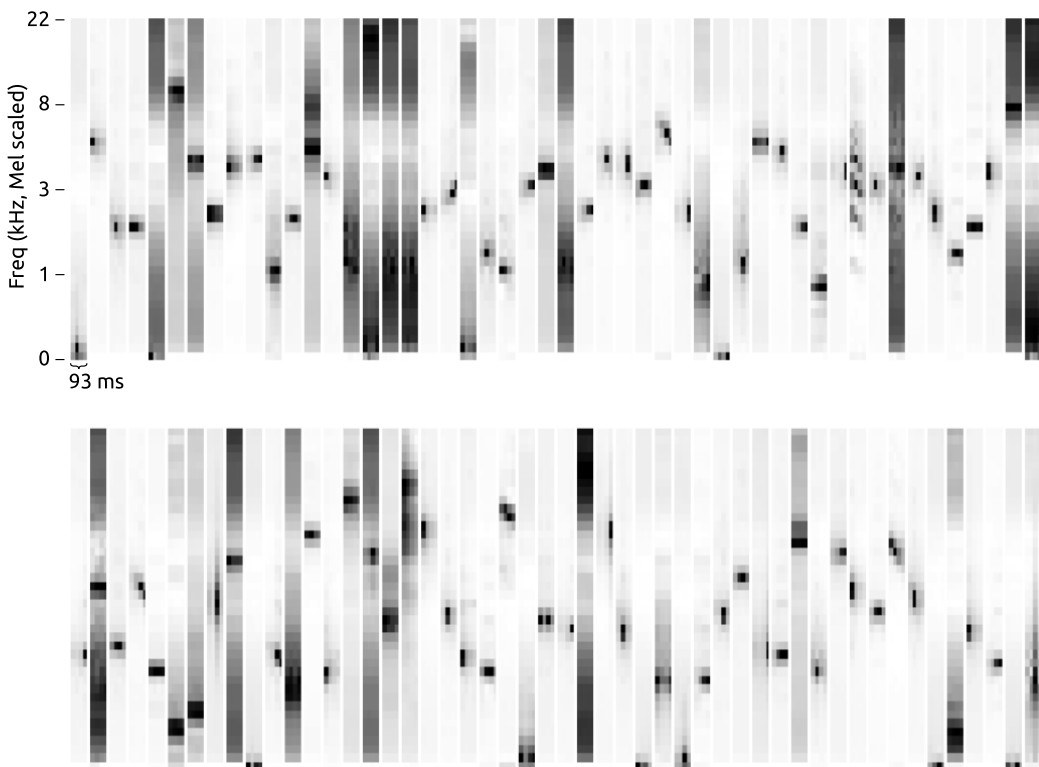

**Figure 10  A random subset of the spectrotemporal bases estimated during one run of feature learning, in this case using 4 frames per base and the *lifeclef2014* dataset.** Each base is visualised as a brief spectrogram excerpt, with dark indicating high values. The frequency axis is nonlinearly (Mel) scaled.

be crucial, since the dataset is the smallest by an order of magnitude in terms of individual labelled items. This issue is compounded by the multilabel scenario, which means there are $2^M$ possibilities for each ground-truth compared against $M$ in the single-label, where $M$ is the number of species considered. Augmenting the training data with *xccoverbl* increased the amount of training annotations, but its failure to boost performance may have been due to differences in kind, since it consisted of single-label recordings of individual birds rather than multilabel recordings of dawn chorus soundscapes. The issue is not due to differences in recording conditions (such as microphone type or recording level): the audio went through various levels of standardisation in our workflow, and caused no problems when added to the feature-learning step only. Instead, it is likely that the single-label annotations are inappropriate for training for the multilabel case. In particular, the single-label constraint may in various cases lead to some bird species being labelled as absent, even though they are audibly present in the background of a recording.

We emphasise that the audio and metadata of the *bldawn* dataset comes directly from a sound archive collection, and the long-form recordings with sparse annotation are exactly the format of a large portion of the holdings in sound archives. Our results (up to around 80% AUC) correspond to a classifier that could provide useful semi-automation of archive labelling (e.g., suggested labels to a human annotator) but not yet a fully automatic process

in that case. This outcome thus reinforces the importance of collecting and publishing annotated audio datasets which fit the intended application. Public collections such as Xeno Canto are highly valuable, but their data do not provide the basis to solve all tasks in species recognition, let alone the other automated tasks (censusing, individual recogntion) that we may wish to perform automatically from audio data.

Our feature-based approach to classification is not the only approach. Template-based methods have some history in the literature, with the main issue of concern being how to match a limited set of templates against the unbounded natural variation in bird sound realisations, in particular the dramatic temporal variability. One technique to compensate for temporal variability is dynamic time warping (DTW) (*Anderson, Dave & Margoliash, 1996*; *Ito & Mori, 1999*) . Recent methods which performed very strongly in the SABIOD-organised contests used templates without any time-warping considerations, making use of a large number of statistics derived from the match between a template and an example (not using just the closeness-of-match) (*Fodor, 2013*) . Other recent methods use templates and DTW but deployed within a kernel-based distance measure, again going beyond a simple one-to-one match (*Damoulas et al., 2010*).

In light of these other perspectives, we note an analogy with our learned features. In the workflow considered here, the new representation is calculated by taking the dot-product between the input data and each base (such as those in Fig. 10), as given in (3). The form of (3) is the same mathematically as spectro-temporal cross-correlation, where the $b_j$ would be thought of more traditionally as "templates". Our features are thus equivalent to the output of an unusual kind of template matching by cross-correlation, where the "templates" are not indivudal audio excerpts but generalisations of features found broadly across audio excerpts, and are also of a fixed short duration (shorter than many song syllables, though long enough to encompass many calls).

A question that arises from this perspective is whether our approach should use longer series of frames, long enough to encompass many types of song syllable entirely. In our tests we found no notable tendency for improved recognition as we increased the number of frames from one to eight, and we also saw many temporally-compact bases learnt (Fig. 10), so we do not believe lengthening the bases is the route to best performance. Further, the advantage of using relatively short durations is that the feature learning method learns *components* of bird vocalisations rather than over-specific whole units. These components may co-occur in a wide variety of bird sounds, in temporally-flexible orders, conferring a combinatorial benefit of broad expressivity. Our two-layer feature learning provides a further level of abstraction over temporal combinations of energy patterns, which is perhaps part of its advantage when applied to our largest dataset. We have not explicitly tested our method in comparison to template-based approaches; the relative merits of such approaches will become clear in further research.

The bases shown in Fig. 10 also bear some likeness with the spectro-temporal receptive fields (STRFs) measured from neurons found in the early auditory system of songbirds (e.g., Fig. 11, adapted from *Hausberger et al. (2000)*). Broadly similar generalisations seem to emerge, including sensitivity to spectrally-compact stationary tones as well as

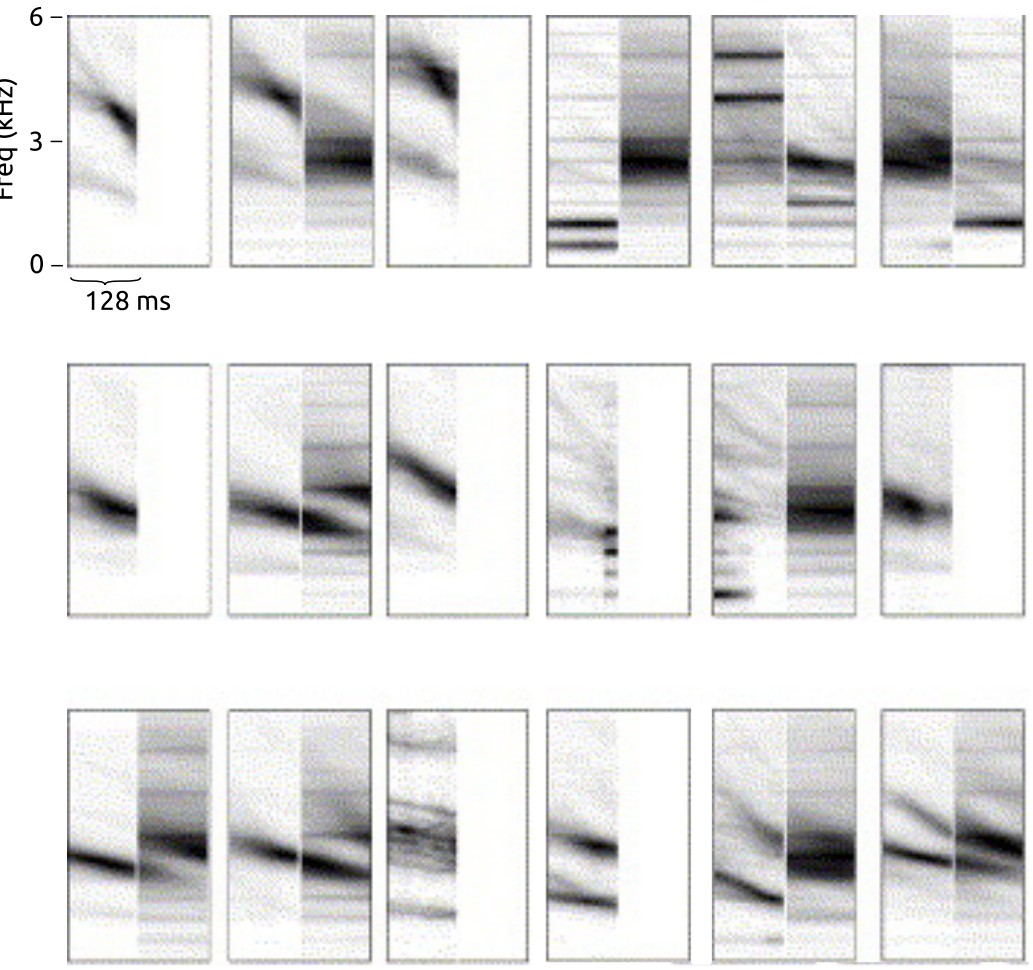

**Figure 11 Spectro-temporal receptive fields (STRFs) measured from individual neurons in auditory field L of starling.** Adapted from *Hausberger et al. (2000)* for comparison with Fig. 10. Each image shows the spectro-temporal patterns that correlate with excitation (left) and inhibition (right) of a single neuron. The frequency axis is linearly scaled.

up-chirps and down-chirps, and combinations of harmonic sounds. We do not make strong claims from this likeness: firstly because our method (spherical k-means) is a simple sparse feature learning method with no designed resemblance to neural processes involved in natural learning, and secondly because STRFs show only a partial summary of the nonlinear response characteristics of neurons. However, sparse feature learning in general is motivated by considerations of the informational and energetic constraints that may have influenced the evolution of neural mechanisms (*Olshausen & Field, 2004*). *Theunissen & Shaevitz (2006)* note that the sensitivities measured from neurons in the avian primary auditory forebrain generally relate not to entire song syllables, but to smaller units which may serve as building blocks for later processing. Biological analogies are not a necessary factor in the power of machine learning methods, but such hints from neurology suggest that the method we have used in this study fits within a paradigm that may be worth further exploration.

## CONCLUSIONS

Current interest in automatic classification of bird sounds is motivated by the practical scientific need to label large volumes of data coming from sources such as remote monitoring stations and sound archives. Unsupervised feature learning is a simple and effective method to boost classification performance by learning spectro-temporal regularities in the data. It does not require training labels or any other side-information, it can be used within any classification workflow, and once trained it imposes negligible extra computational effort on the classifier. In experiments it learns regularities in bird vocalisation data with similar qualities to the sensitivities of bird audition reported by others.

The principal practical issue with unsupervised feature learning is that it requires large data volumes to be effective, as confirmed in our tests. However, this exhibits a synergy with the large data volumes that are increasingly becoming standard. For our largest dataset, feature learning led to classification performance up to 85.4% AUC, whereas without feature learning the performance peaked at 82.2% for raw Mel spectra or 69.3% for MFCCs.

In our tests, the choice of feature set made a much larger difference to classification performance than any of our other configuration choices (such as the use of noise reduction, decision pooling, or binary relevance). Although MFCCs have been widespread as baseline features for bird species recognition, the undigested Mel spectra themselves may often be more appropriate for benchmarking, since they dramatically outperform MFCCs in most of our tests. We recommend that researchers should benchmark future sound representations against both MFCCs and raw (Mel) spectra. Across our various tests in single-label and multilabel settings, unsupervised feature learning together with a multilabel classifier achieved peak or near-peak classification quality.

This study, thanks to the large-scale data made available by others, has demonstrated strong performance on bird sound classification is possible at very large scale, when the synergy between big data volumes and feature learning is exploited. However, automatic classification is not yet trivial across all domains, as demonstrated by the lack of improvement on our *bldawn* dataset of dawn chorus recordings. The research community will benefit most from the creation/publication of large bird audio collections, labelled or at least part-labelled, and published under open data licences.

## ACKNOWLEDGEMENTS

We would like to thank the people and projects which made available the data used for this research: the SABIOD research project (led by Prof Glotin, and part of MI CNRS MASTODONS) which co-organised the NIPS4B and LifeCLEF challenges; the Xeno Canto website and its many volunteer contributors; the Biotope society; the British Library Sound Archive and its contributors, and curator Cheryl Tipp.

### Funding

This work was supported by EPSRC Leadership Fellowship EP/G007144/1 and EPSRC Early Career Fellowship EP/L020505/1. The funders had no role in study design, data collection and analysis, decision to publish, or preparation of the manuscript.

### Grant Disclosures

The following grant information was disclosed by the authors:
EPSRC: EP/G007144/1, EP/L020505/1.

### Competing Interests

The authors declare there are no competing interests.

### Author Contributions

- Dan Stowell conceived and designed the experiments, performed the experiments, analyzed the data, contributed reagents/materials/analysis tools, wrote the paper, prepared figures and/or tables, reviewed drafts of the paper.
- Mark D. Plumbley conceived and designed the experiments, reviewed drafts of the paper.

### Data Deposition

The following information was supplied regarding the deposition of related data:

- The *xccoverbl* data is archived at https://archive.org/details/xccoverbl_2014—it is composed of sound files sourced from http://www.xeno-canto.org/. The sound file authors and other metadata are listed in a CSV file included in the dataset.
- The *bldawn* dataset is available on request from the British Library Sound Archive (BLSA). Our machine-readable version of the species metadata is downloadable from Figshare, and lists the file identifiers corresponding to the BLSA records: http://dx.doi.org/10.6084/m9.figshare.1024549.
- The *nips4b* dataset is downloadable from the SABIOD/nips4b website: http://sabiod.univ-tln.fr/nips4b/challenge1.html.
- The *lifeclef2014* dataset is available from the Lifeclef website: http://www.imageclef.org/2014/lifeclef/bird.

### Supplemental Information

Supplemental information for this article can be found online at http://dx.doi.org/10.7717/peerj.488.

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
