# Peer review of "Automatic large-scale classification of bird sounds is strongly improved by unsupervised feature learning"

_PeerJ, doi:10.7717/peerj.488_

## Round 0.1 · original submission · Minor Revisions

Two Reviewers have provided their comments about this manuscript. If you wish to revise your manuscript, please take the referee comments fully into account and provide point-by-point responses with a full list of changes.

Reviewer 1 ·

Basic reporting

The article is well written using clear and unambiguous text. It includes a good introduction although some studies regarding bird classification are only given implicit, citing a survey written by the authors a few years ago. Sufficient background information is provided demonstrating how the work fits into the field of knowledge. The spherical k-means algorithm for feature learning is provided as Python code and made available as supplementary information.
Some additional figures would have been helpful, e.g. one representative spectrogram per dataset to get an impression of the differences and/or a before and after spectrogram to see the influence of the noise reduction applied.
The in-text citations should be checked again for consistent formatting.

Experimental design

A wide range of different methods, parameter variations and classifier settings are compared regarding classification performance on different datasets. The methods are clearly described and the variations are logical, well explained and justified.
For the replication of results it would be helpful to know the random forest parameter used for training (number of estimators, max_features, min_sample_split) for the single- and multi-label resp. binary relevance approach.

Validity of the findings

The study is relevant and provides valuable information for researchers working in the same field. Unfortunately some results are not consisting over all datasets, e.g. the boost of classification performance regarding the use of feature learning for the bldawn dataset. The authors try to give reasons for this but a more detailed explanation would be desirable.

Comments for the author

I have the following concerns:

Abstract:

- “We experimentally compare … with a random forest classifier.” -> should be rephrased

- “MFCCs are of limited power” -> statement is to overgeneralized and should be put into perspective

Remark: MFCCs worked quite well (with appropriate modifications) in the NIPS 2013 Bird Classification Challenge considering the results of the second best team (les bricoleurs: e.g. Matt Wescott), I don’t know if they published any paper about it, if yes, it should be mentioned (not necessarily in the abstract)


Text:

- Line 55: “acoustically/perceputally” -> perceptually (right spelling!) is enough
- Line 63: “approximately logarithmic sensitivity of human hearing” -> relatedness to pitch missing (could also refer to log. sensitivity regarding perception of loudness)
- Line 137: "in hundreds of individual crossvalidated classification tests" -> sounds vague, can you be more specific or give the exact number
- Line 150: "the lifeclef2014 dataset is single-label” -> it should be mentioned somewhere that it could also be used as multi-label, if you consider the metadata information about background species
- Line 164-171: the name of dataset (xccoverbl) should be mentioned within the paragraph
- 185-186: text without line numbers
- Line “For this we use a method that has has shown promise” -> only one “has”
- Line 222: “windowing and no overlap” -> quite uncommon choice, is there a reason?
- Line 222: “We filtered out spectral energy below 500 Hz, a common choice …” -> please explain why common or cite studies that use the same cutoff frequency

Remark: some species are vocalizing below 500 Hz: (e.g. Botaurus stellaris, Bubo bubo)

- figures (with box plot) -> please specify what is represented by the ends of the whiskers
- Line 428: “result attained 91.8%” -> please add citation
- Line 541: "MFCCs cannot be recommended ..." -> same concern as in Abstract

Reviewer 2 ·

Basic reporting

No comments

Experimental design

No Comments

Validity of the findings

No Comments

Comments for the author

The paper is generally thorough and solid. New machine learning methods were used to help classify bird sounds in an unsupervised way. Multiple results were presented and showed the effectiveness of the methods. In my perspective, only some minor explanations and clarifications may need to be added or adjusted.

1. I would like to see more reasons on choosing spherical k-means over other methods on feature learning, especially the methods like deep neural network as author mentioned.

2. The four datasets used in the paper, one of which has only 60 items. And only two of them can be combined together. So why not choose some other (or even more common) datasets?

3. In section of classification method, I suggest a general overview of random forest classifier with the selection of the parameter. As authors mentioned, no manual adjustment is made. But it would be more clear if the default values of the parameters are included.

4. Another concern related to section above is that without parameter tuning, how do you avoid under or over fitting? Or simply why not do a parameter tuning for better results?

5. In line 306, GLM should be referred as generalized linear model.

6.In line 380, the high score found when tested on training data for bldawn. Is that due to the property of the tree bagging methods or the reason as authors explained? Since the sample size is relatively small for this dataset, the bagging methods may result in an accurate prediction of training data.

7. When comparing results and drawing conclusions, is there previous accuracy results on the datasets to include and compare with?

---

## Round 0.2 · accepted · Accept

The authors had made great progresses according to the reviewers' comments. Hence, the conditional acceptation (after revising a few minor issues pointed out by reviewer 1) is my decision now.

Reviewer 1 ·

Basic reporting

ok

Experimental design

ok

Validity of the findings

ok

Comments for the author

The authors followed most of the reviewers' suggestions and added clarifying comments. After addressing a few minor issues (see below) the paper should be accepted for publication in PeerJ.

Minor issues:

- Line 55: "Raw audio data is not generally suitable input to a classification algorithm..."
--> seems to be a word missing

- Line 198: "This leads to a preference for techniques of low computational complexity, and which can be applied..."
--> maybe rephrase (grammar)

- Line 452: "...the winning result attained 91.8% (Glotin et al., 2013)..."
--> please cite the specific paper you are refering to in the proceedings (author, title, pages), not only the conference proceedings and editors (format examples: http://guides.library.vu.edu.au/content.php?pid=270421&sid=2230822 or https://peerj.com/about/author-instructions/#reference-section)

Reviewer 2 ·

Basic reporting

No comments

Experimental design

No comments

Validity of the findings

No comments

Comments for the author

The paper is very organized and solid with plenty of results and discussion. Congratulations to the authors for a great work.

Sufficient backgrounds and introduction are provided. Multiple effective classification and evaluation methods are used and clearly illustrated. Promising results are obtained with detailed discussion.